# Uncovering the Burden of Rhinitis in Patients Purchasing Nonprescription Short-Acting β-Agonist (SABA) in the Community

**DOI:** 10.3390/pharmacy11040115

**Published:** 2023-07-10

**Authors:** Sara Alamyar, Elizabeth Azzi, Pamela Srour-Alphonse, Rachel House, Biljana Cvetkovski, Vicky Kritikos, Sinthia Bosnic-Anticevich

**Affiliations:** 1Sydney Medical School, The University of Sydney, Sydney, NSW 2006, Australia; sala6507@uni.sydney.edu.au; 2Quality Use of Respiratory Medicine Group, Woolcock Institute of Medical Research, The University of Sydney, Sydney, NSW 2006, Australia; elizabeth.a.azzi@gsk.com (E.A.); pamela@alphonse.com.au (P.S.-A.); biljana.cvetkovski@woolcock.org.au (B.C.); vicky.kritikos@woolcock.org.au (V.K.); sinthia.bosnic-anticevich@woolcock.org.au (S.B.-A.); 3GSK Australia, Pty., Ltd., Ermington, NSW 2115, Australia; 4Sydney Pharmacy School, The University of Sydney, Sydney, NSW 2006, Australia; 5Sydney Local Health District, Sydney, NSW 2050, Australia

**Keywords:** allergic rhinitis, asthma, attitudes, behaviors, medication, perception, rhinitis

## Abstract

Asthma and rhinitis are common comorbidities that amplify the burden of each disease. They are both characterized by poor symptom control, low adherence to clinical management guidelines, and high levels of patient self-management. Therefore, this study aims to investigate the prevalence of self-reported rhinitis symptoms in people with asthma purchasing Short-Acting Beta Agonist (SABA) reliever medication from a community pharmacy and compare the medication-related behavioral characteristics among those who self-report rhinitis symptoms and those who do not. Data were analyzed from 333 people with asthma who visited one of eighteen community pharmacies in New South Wales from 2017–2018 to purchase SABA and completed a self-administered questionnaire. Participants who reported rhinitis symptoms (71%), compared to those who did not, were significantly more likely to have coexisting gastroesophageal reflux disease (GERD), overuse SABA, and experience side effects. They may have been prescribed daily preventer medication but forget to take it, and worry about its side effects. They were also more likely to experience moderate-to-severe rhinitis (74.0%), inaccurately perceive their asthma as well-controlled (50.0% self-determined vs. 14.8% clinical-guideline defined), and unlikely to use rhinitis medications (26.2%) or daily preventer medication (26.7%). These findings enhance our understanding of this cohort and allow us to identify interventions to improve patient outcomes.

## 1. Introduction

Rhinitis, characterized by an inflammation of the nasal mucosa, can be generally classified as allergic rhinitis (AR) and non-allergic rhinitis (NAR) [1]. In some cases, however, a person can have components of both AR and NAR, and this is known as mixed rhinitis [2]. Differentiating between the different types of rhinitis can be difficult because the diagnostic criteria for various forms are not always clear-cut. While some medications may be effective for AR (antihistamines and intranasal corticosteroids), they may be less effective for other forms of rhinitis. Recent data suggest that 10–40% of patients have AR and 6–25% have NAR [3,4].

AR, the most common form of rhinitis, is associated with an IgE-mediated response against allergens [5]. AR affects an estimated 19% of Australians and up to 40% of the global population [3,4,6,7]. Although often dismissed as merely a conglomeration of nasal symptoms and not ‘life-threatening,’ AR can impact quality of life, cognitive function, mental and ocular health, and sleep [5,8]. Despite its significant burden on individuals and the healthcare system [9], it remains largely under-diagnosed, under-treated and under-researched [1,6,10,11].

Rhinitis, both allergic and non-allergic, is a risk factor for developing asthma [5,9,12] and a common comorbidity; studies have found that 15–38% of people with AR also have asthma, and 74–81% of people with asthma report symptoms of rhinitis [6,13,14,15,16]. Asthma mirrors AR with persistently high levels of poor control in the community [17,18,19,20,21]. Poorly treated AR is associated with various poor asthma outcomes, which include increased asthma exacerbations [5], poor asthma control [13], and increased use of asthma medication and related healthcare resources [13]. A classic example of the effect of AR on people with asthma was seen in its association with the Thunderstorm Asthma Event in Australia, when a sudden surge of humidity provoked osmotic shock, breaking up pollen grains into small, inhalable particles [22,23]. Of the patients who attended the nearby hospital emergency department after the incident, 87% had prior AR without an asthma diagnosis [22,23].

Both asthma and rhinitis are largely managed in the community, with a significant degree of self-management by individuals whose perceptions of their symptoms and behaviors often do not reflect clinical management guidelines [19,24,25,26,27]. Furthermore, while guidelines recommend managing rhinitis alongside asthma, this is incredibly challenging because of the high rates of under-reporting and under-diagnosis of rhinitis in practice [17,24,28,29]. Patients often self-diagnose and self-medicate for AR from community pharmacies without the intervention of a healthcare professional [24]. In a study by Nolte et al., 32% of patients with rhinitis did not have a doctor’s diagnosis of their condition, and 83% of patients with moderate-to-severe AR were undertreated [29]. Accordingly, there is a lack of knowledge about this comorbid population and a need for more research that reflects real-world circumstances [19,30]. An improved understanding of the characteristics, attitudes, and behavior of individuals with comorbid asthma and rhinitis in the community offers the potential to improve their current poor-management outcomes [13,30,31].

Currently in Australia, many visit their community pharmacy for advice and medications, providing a unique setting and perspective to know people with both asthma and rhinitis and their medication-taking behaviors. Short-Acting Beta Agonist (SABA) medication can be purchased without a prescription, for it is regulated as “Pharmacist Only Medicine” in Australia, meaning SABA is deemed safe to use with a pharmacist’s consent but without a doctor’s prescription. The overuse of SABA medication may indicate poor asthma control and adverse health outcomes as this medication does not treat the underlying inflammatory pathology that gives rise to asthma symptoms [32,33]. People who overuse SABA medication have also been shown to have bothersome rhinitis symptoms [34]. It is therefore hypothesized that people with both asthma and rhinitis have poor medication-taking behavior, which subsequently impacts both their rhinitis and asthma control. This study has two aims: to investigate the prevalence of self-reported rhinitis symptoms in people with asthma who purchase SABA without a prescription from a community pharmacy; and to compare the medication-related behavioral characteristics of those who self-report rhinitis symptoms and those who do not.

## 2. Materials and Methods

### 2.1. Study Design and Data Source

A cross-sectional study design was employed to investigate patients purchasing nonprescription SABA medication in community pharmacies [32,35]. Data were collected from October 2017 to October 2018. These consisted of anonymized responses from participants who consented to answer a structured, self-administered questionnaire in a pharmacy (Appendix A, Figure A1). The data were collected from 18 pharmacies across various regions in New South Wales, Australia. 

### 2.2. Inclusion Criteria

Pharmacy customers were invited to complete the questionnaire if they were aged 16 years and over, purchasing nonprescription SABA medication for themselves, and able to complete the questionnaires in English. There were no exclusion criteria.

### 2.3. Data Collection

The questionnaire was developed based on validated tools derived from asthma guidelines [21], AR guidelines [1,36], and empirical evidence [17,37], which have been described in detail by Azzi et al., 2019 [32]. The questionnaire was designed to facilitate quick and easy self-administration and was reviewed by an expert panel of clinical pharmacists, general practitioners, and respiratory specialists. Anonymized data were collected from all participants and included demographic characteristics, comorbidities, SABA medication use, related side effects, and self-reported rhinitis symptoms.

Self-reported rhinitis symptoms were assessed with the question, “Do you have either an: itchy, runny, blocked nose or sneezing when you don’t have a cold?” This question was derived from the definition of rhinitis in “Allergic Rhinitis and its Impact on Asthma and IPCRG” [1,36]; see Bosnic-Anticevich et al. [17]. The severity of the rhinitis is determined by how “bothersome” patients’ symptoms are in affecting their sleep, performance, or other daily activities. Patients who experienced “bothersome” symptoms were classified as having moderate-to-severe rhinitis [17].

Participants who reported an asthma diagnosis were then invited to provide further information about their asthma control, asthma medication-related behaviors, attitudes toward and beliefs about asthma, occurrence of exacerbations, and healthcare utilization. Asthma control was evaluated based on criteria from the Global Initiative for Asthma (GINA) guidelines and was categorized as well-controlled, partly controlled, or uncontrolled [21]. The participants’ perceived control was dichotomised into controlled (well/completely controlled) and not controlled (somewhat/poorly/not controlled at all). Attitudes toward and beliefs about asthma were assessed using the “Asthma Patient Profiling Tool”, where participants had to agree or disagree with 10 attitudinal statements relating to these areas across 8 summary factors [37].

### 2.4. Data Analysis

Data were analysed using SPSS version 28 (SPSS-IBM, Chicago, IL, USA). Descriptive statistics were used to summarize participants’ demographic and baseline data from the subpopulation of participants with reported asthma diagnoses. Categorical variables were analysed using the Pearson Chi-square test, and continuous variables were analysed using the independent sample *t*-test with a significance value of *p* < 0.05.

## 3. Results

### 3.1. Participant Characteristics

Of the 412 participants recruited, 80.2% (333/412) had an asthma diagnosis (Figure 1), of whom 70.8% (236/333) reported rhinitis symptoms (“Do you have either an: itchy, runny, blocked nose or sneezing when you don’t have a cold?”).

Table 1 summarizes participants’ demographic characteristics and comorbidities. Participants with self-reported rhinitis symptoms had a mean (SD) age of 43.4 (18.2) years; 54.1% were female, 83.2% had good-excellent overall health, and 35.1% were current/past smokers. Almost two-thirds (74.6%, 176/236) of the participants were classified with moderate-to-severe rhinitis with 26.7% (63/236) using tablets or nasal sprays to treat it and 26.2% (39/236) reporting that their rhinitis medications did not alleviate their symptoms. The participants reporting rhinitis symptoms were more likely to be female than male and reported having gastroesophageal reflux disease (GERD) as a comorbid condition. Of the 200 patients having comorbid hay fever, only 7.5% (15/200) reported no rhinitis symptoms.

### 3.2. Asthma Control, Perception and Healthcare Utilisation

Overall, 82.9% of participants had GINA-defined partly controlled and uncontrolled asthma, and based on patients’ perception, only 46.5% perceived their asthma as not controlled (i.e., somewhat/poor/not controlled at all). A greater proportion of participants reporting rhinitis symptoms had partly controlled or uncontrolled asthma (85.2% vs. 77.3%) and perceived their asthma as not controlled (50.0% vs. 38.1%) compared to those who did not report rhinitis symptoms. Half (50.0%, 118/236) of the participants with asthma who reported rhinitis symptoms perceived their asthma as controlled, and only 14.8% fulfilled the GINA criteria for well-controlled asthma (Table 2).

In the 12 months prior to completing the questionnaire, 72.1% of the participants reported seeing a physician for their asthma, 22.2% had used oral corticosteroids for their asthma, and 11.7% had visited an emergency department or been hospitalized for their asthma (Table 2). Thirty-four percent (113/333) of the participants reported having an asthma action plan, and 72.4% reported receiving previous asthma education (Table 2). No significant differences emerged with respect to their asthma control, perception, and healthcare utilization between those reporting rhinitis symptoms and those who did not, as shown in Table 2.

### 3.3. Asthma Medication-Related Behaviors

Table 3 summarizes participants’ medication-related behaviors. Overall, 57.7% of participants with asthma were classified as having overused SABA medication in the past week (defined as using SABA more than three times per week), with 43.5% who reported experiencing one or more SABA-related side effects (Table 3). Participants reporting rhinitis symptoms were more likely to overuse SABA medications (60.2% vs. 51.5%), experience SABA side effects (48.3% vs. 32.0%, *p* = 0.007), particularly palpitations (11.9% vs. 3.1%, *p* = 0.012), and owned a greater number of SABA inhalers than others (2.5 ± 1.15 vs. 1.98 ± 1.0, *p* = 0.001) (Table 3).

In total, 70.0% of the participants currently owned a preventer medication. While 60.7% of participants reported being instructed to use a preventer “every day”, only 27.3% did so (Table 3). Overall, 27.3% of participants reported being instructed to use a preventer “as required”, with 24.3% reporting “I take it only when have symptoms” (Table 3). Patients reporting rhinitis symptoms were more likely to have been instructed to use their preventer medication daily compared to those who did not report rhinitis symptoms (63.6% vs. 53.6%, *p* = 0.022), but there was no significant difference in the rates of daily preventer use between the two groups (Table 3). The most common reason for failing to use preventers daily among participants with asthma was the belief that they “don’t need it” (61.0%, 203/333), “forget to take it” (45.6%, 152/333) and “worried about side effects” (20.1%, 67/333). Participants reporting rhinitis symptoms were more likely to forget to take their preventers (51.7% vs. 30.9%, *p* = 0.001) and be concerned about side effects (22.9% vs. 13.4%, *p* = 0.05) than those reporting no rhinitis symptoms (Table 3).

### 3.4. Attitudes and Perceptions towards Asthma

Table 4 summarizes participants’ attitudes and perception about asthma. Overall, 70.9% (236/333) saw themselves as healthy and fit, 54.7% (182/333) claimed that their symptoms are not serious, and 45.6% (152/333) were worried about what their asthma would be like in 10 years.

As stated in Table 4, those reporting rhinitis symptoms were more likely to agree with the statement, “I have no time to think about my health as other things are more important” (37.3% vs. 19.6%, *p* = 0.002), and less likely to agree that “my symptoms are not serious” (50.0% vs. 66.0%, *p* = 0.008). There were no significant differences between those reporting rhinitis symptoms and those who did not (Table 4).

## 4. Discussion

This study further enhanced the perception, burden, and management of people with asthma and rhinitis in a real-world setting. This study among people with asthma purchasing SABA without a prescription from community pharmacies in Australia reveals that while only 17.1% had GINA-defined controlled asthma, over two-thirds reported concomitant moderate-to-severe rhinitis. People with asthma and self-reported rhinitis symptoms were significantly more likely to (1) have coexisting GERD; (2) overuse SABA, experience side effects, and own a greater number of SABA inhalers at any one time; (3) be prescribed daily preventer medication but also have concerns about side effects and forget to use it; and (4) be inclined to feel as if they have no time to think about their health and claim that their asthma symptoms are not serious. Further, a large proportion of these participants held inaccurate perceptions about their level of asthma control, had moderate-to-severe rhinitis, and were unlikely to use medication to treat their rhinitis. These insights gained from this study can be employed to build targeted interventions that might potentially be able to improve outcomes for patients affected by both asthma and rhinitis.

A significant proportion of participants who self-reported rhinitis symptoms also had coexisting GERD. The association between AR and GERD is not commonly reported but has been explored in recent studies. Kung et al., 2019 identified AR as a risk factor for GERD [38]. It should also be noted that laryngopharyngeal reflux results from GERD that reaches the upper airway respiratory tract [39]. Therefore, the increase in GERD symptoms among our participants with comorbid AR and asthma is not surprising. This requires further investigation in the pharmacy setting.

Participants in this study struggled with the burden of their co-morbidity; as a result, the management of both their asthma and rhinitis was sub-optimal. Only 14.8% of those reporting rhinitis symptoms had well-controlled asthma. Participants also had similarly poor control of their comorbid AR, with three-quarters characterized as having moderate-to-severe symptoms. In line with existing research, both conditions in these individuals appeared significantly self-managed, were inconsistent with current guidelines, and indicated a lack of sufficient education about both conditions and their relationship [10,17,40]. Although the cohort of participants purchasing SABA reliever medication without a prescription may have introduced bias to the sample, more than half (57.7%) were overusing SABA medications, and those with rhinitis symptoms had a greater number of SABA inhalers at any one time than those who did not report rhinitis symptoms. Furthermore, almost half (48.3%) of the participants who reported rhinitis symptoms also reported SABA-related side effects, had low utilization of preventer medication, and had no asthma action plans. Although 70.0% had a preventer, barely one-quarter of those with asthma and rhinitis symptoms reported taking their preventer every day. These individuals were more likely to report forgetting to take their preventer and being concerned about the side effects. Almost two-thirds (63.1%) thought that their preventer was not needed. The high proportion of SABA overuse and non-adherence to preventer therapy is concerning, particularly since there is growing evidence that the former is associated with increased asthma morbidity and mortality [41,42]. The extent to which paradoxes regarding the previous GINA stepwise management of asthma could contribute to SABA overuse and preventer underuse has also been recognized [40,43,44]. Interestingly, this prioritization of medication by seeking rapid symptom relief is mirrored in studies of patient preferences for medication in AR [13]. Barely one-third (34.3%) of participants reporting rhinitis symptoms also reported having an asthma action plan. Half considered their asthma symptoms not serious, even though these perceptions were probably inaccurate and their asthma was in fact poorly controlled. This mirrors research in AR about inaccurate patient perceptions of the severity of their symptoms [11,45]. While a majority (72.4%) of participants in this study reported some previous asthma education and 72.1% had visited their doctor for asthma in the past 12 months, these results suggest further education and more active management are worthy targets for intervention.

Although their rhinitis symptoms were moderate-to-severe, only a quarter of the participants reported taking medication to treat their rhinitis symptoms, with a quarter of this cohort finding the medications they were taking to be ineffective. Unfortunately, data were not collected about which type of rhinitis medications the participants were taking. Furthermore, over one-fifth of those reporting rhinitis symptoms did not report having “hay fever,” which may suggest that they are either undiagnosed, unaware that their symptoms reflect AR, or may even have NAR because hay fever is synonymous with seasonal allergies. While such findings are consistent with previous research regarding the underdiagnosed and under-treated nature of rhinitis [10,24,25,45], this research highlights the need to increase patient education about rhinitis and set a target for optimizing control of both asthma and rhinitis. For example, intranasal corticosteroids, the first-line treatment for AR [46], have been found to improve asthma outcomes in patients with comorbid asthma and AR [47]. Given the above findings, it is unsurprising these individuals were also likely to feel as if they have no time to think about their health, an attitude likely to contribute to sub-optimal management of both conditions.

High levels of self-management, inaccurate perceptions, and poor disease control provide both challenges and impetus for engaging individuals with comorbid asthma and rhinitis. Community pharmacists appear particularly well placed to take advantage of this opportunity [24,25,45]. Asthma and rhinitis are chronic conditions likely to evolve over an individual’s lifetime and require ongoing education and management [25,48,49]. There is an increasing plethora of medications, many available without a prescription, and guidelines regarding the management of both conditions are constantly revised [24,25]. For example, GINA’s 2021 recommendations now advise against SABA monotherapy as a first-line management for asthma and emphasise the role of low-dose inhaled corticosteroids [41]. Translating these changes into individual practice could be challenging, given attitudes and behaviors surrounding the preference for SABA and the inconsistent use of preventer medication. As seen in other studies [17,24], it is likely that these individuals are unaware of the links between their rhinitis and asthma symptoms and the importance of treating rhinitis, both for symptomatic relief in its own right but also as a trigger for their asthma. Given the characteristics of those attending community pharmacies to purchase nonprescription SABA medication, the findings of this study indicate there is value in screening this population for rhinitis symptoms and determining whether these are being treated. Furthermore, individuals need to be aware of asthma symptoms and develop an asthma management plan, which includes a knowledgeable use of SABA and preventer medications. Appropriately trained and resourced pharmacists could play a crucial role in engaging with their customers, educating them about their health challenges, and thereby improving outcomes. While the precise form of intervention is a topic for further research, there is growing evidence that consistent education from a multidisciplinary healthcare team contributes to easing the burden of asthma on both the individual and the healthcare system [19,48,49].

This study benefits from real-world evidence from a community cohort of varying socio-demographic areas within New South Wales and a detailed questionnaire with both quantitative and qualitative data. Limitations include the relatively small sampling confined to one state, which may reduce the general application of results and the use of patient-reported outcomes with a potential for inaccuracy, memory bias; and finally, lack of medical confirmation for reported asthma or rhinitis diagnoses. Moreover, it is difficult to distinguish between AR and other forms of rhinitis without a proper diagnosis [8,45].

## 5. Conclusions

Rhinitis is a common comorbidity of asthma, and this study reinforces that these conditions are often trivialized and characterized by poor self-management, non-adherence to clinical guidelines, and lack of symptom control. Together, they contribute to a higher burden of disease for the individual and the healthcare system. Overreliance on SABA medication and under-utilization of preventer and rhinitis medications are underpinned by inaccurate individual perceptions and a lack of consistent patient education. However, as this study highlights, this synergy between comorbidities also offers opportunities for intervention to improve outcomes.

## Figures and Tables

**Figure 1 pharmacy-11-00115-f001:**
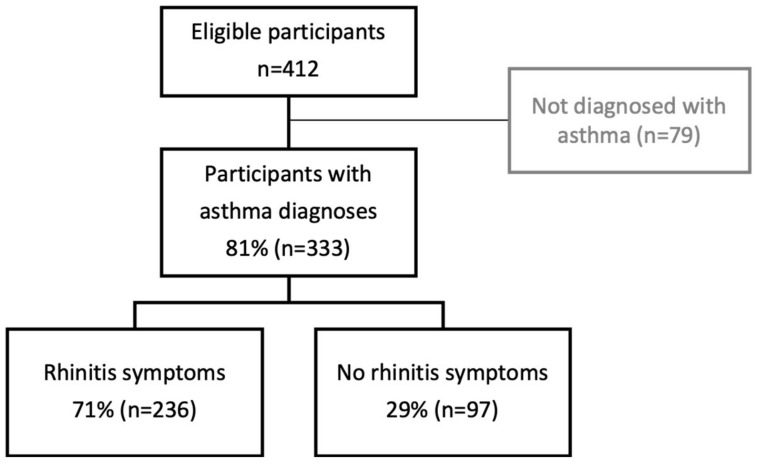
Consort diagram of participants.

**Table 1 pharmacy-11-00115-t001:** Participants’ characteristics (n = 333).

Characteristic	Overall (n = 333)	No Rhinitis Symptoms (n = 97)	Rhinitis Symptoms (n = 236)	*p* Value *
Age, mean ± SD	43.4 ± 18.2	43.9 ± 19.3	43.3 ± 17.7	0.779
**Gender, n (%)**				
Female	180 (54.1)	43 (44.3)	137 (58.1)	**0.029**
**Overall Health, n (%)**				
Fair/Poor	56 (16.8)	15 (15.5)	41 (17.4)	0.749
Excellent/Very Good/Good	277 (83.2)	82 (84.5)	195 (82.6)	
**Comorbidities, n (%)**				
Hay fever ^^^	200 (60.1)	15 (15.5)	185 (78.4)	**<0.001**
Eczema	67 (20.1)	13 (13.4)	54 (22.9)	0.052
Gastroesophageal reflux disease	58 (17.4)	10 (10.3)	48 (20.3)	**0.038**
Depression	41 (12.3)	8 (8.2)	33 (14.0)	0.198
Obstructive sleep apnea	31 (9.3)	6 (6.2)	25 (10.6)	0.299
Cardiac disease	30 (9.0)	9 (9.3)	21 (8.9)	1.000
Obesity	29 (8.7)	11 (11.3)	18 (7.6)	0.289
**Smoking history, n (%)**				
Never smoked	216 (64.9)	66 (68.0)	150 (63.6)	0.452
Current/Past Smoker	117 (35.1)	31 (32.0)	86 (36.4)	

^^^ based on participants self-reporting hay fever, * No rhinitis symptoms versus rhinitis symptoms.

**Table 2 pharmacy-11-00115-t002:** Asthma control, perceptions, and healthcare utilization (n = 333).

	Overall (n = 333)	No Rhinitis Symptoms (n = 97)	Rhinitis Symptoms (n = 236)	*p* Value *
**Asthma symptom control (GINA-defined) in the previous 4 weeks, n (%)**				
Well-controlled	57 (17.1)	22 (22.7)	35 (14.8)	0.108
Partly controlled and uncontrolled	276 (82.9)	75 (77.3)	201 (85.2)	
**Perceived control in the past 4 weeks, n (%)**				
Controlled	178 (53.5)	60 (61.9)	118 (50.0)	0.054
Not controlled	155 (46.5)	37 (38.1)	118 (50.0)	
**Acute exacerbations in the past 12 months, n (%)**				
Seen a doctor about asthma ≥1	240 (72.1)	70 (72.2)	170 (72.0)	1.000
Use oral corticosteroids for asthma ≥1	74 (22.2)	17 (17.5)	57 (24.2)	0.200
Been to emergency department or hospitalized for asthma ≥1	39 (11.7)	13 (13.4)	26 (11.0)	0.575
**Asthma Action Plans, n (%)**				
Patient with an asthma action plan	113 (33.9)	32 (33.0)	81 (34.3)	0.899
Patient received previous asthma education	241 (72.4)	71 (73.2)	170 (72.0)	0.893

Abbreviations: GINA, Global Initiative for Asthma; * No rhinitis symptoms versus rhinitis symptoms.

**Table 3 pharmacy-11-00115-t003:** Asthma medication-related behaviors (n = 333).

	Overall (n = 333)	No Rhinitis Symptoms (n = 97)	Rhinitis Symptoms (n = 236)	*p* Value *
**SABA**				
**SABA use in the last week, n (%)**				
≥3 times per week	192 (57.7)	50 (51.5)	142 (60.2)	0.179
**Highest number of puffs used in one day, in the last 4 weeks, n (%)**				
1–4 puffs	132 (39.6)	41 (42.3)	91 (38.6)	0.540
5 puffs or more	201 (60.4)	56 (57.7)	145 (61.4)	
**SABA inhalers at any one time, mean ± SD**	2.35 ± 1.1	1.98 ± 1.0	2.5 ± 1.15	0.001
**SABA safe to use, n (%)**				
Yes	266 (79.9)	83 (85.6)	183 (77.5)	0.101
**SABA side effects experienced, n (%)**				
Yes	145 (43.5)	31 (32.0)	114 (48.3)	0.007
**Type of SABA side effects, n (%)**				
Dry mouth	53 (15.9)	12 (12.4)	41 (17.4)	0.323 ^a^
Palpitations	31 (9.3)	3 (3.1)	28 (11.9)	**0.012** ^a^
Tremor	25 (7.5)	5 (5.2)	20 (8.5)	0.365 ^a^
Chest tightness	10 (3.0)	3 (3.1)	7 (3.0)	1.000 ^a^
Muscle cramps	6 (1.8)	1 (1.0)	5 (2.1)	0.676 ^a^
Headache	11 (3.3)	2 (2.1)	9 (3.8)	0.520 ^a^
**PREVENTER**				
**Currently own a preventer medication, n (%)**				
Yes	233 (70.0)	66 (68.0)	167 (70.8)	0.693
**Worry about long-term effects of preventer, n (%)**				
Yes	164 (51.4)	53 (58.2)	111 (48.5)	0.137
No	156 (48.8)	38 (41.8)	118 (51.5)	
**Preventer instruction**				
Every day	202 (60.7)	52 (53.6)	150 (63.6)	0.022
As required	91 (27.3)	26 (26.8)	65 (27.5)	
Don’t know	40 (12.0)	19 (19.6)	21 (8.9)	
**Preventer adherence, n (%)**				
I take it every day	91 (27.3)	28 (28.9)	63 (26.7)	0.072
I take it some days but not on others	84 (25.2)	27 (27.8)	57 (24.2)	
I used to take it but now do not	29 (8.7)	6 (6.2)	23 (9.7)	
I take it only when I have symptoms	81 (24.3)	16 (16.5)	65 (27.5)	
I never take it	48 (14.4)	20 (20.6)	28 (11.9)	
**Reasons for not using preventers daily ^a^, n (%)**				
Don’t need it	203 (61.0)	54 (55.7)	149 (63.1)	0.200 ^a^
Worried about side effects	67 (20.1)	13 (13.4)	54 (22.9)	0.050 ^a^
Don’t like it	23 (6.9)	6 (6.2)	17 (7.2)	0.700 ^a^
It does not work	50 (15.0)	9 (9.3)	41 (17.4)	0.060 ^a^
Forget to take it	152 (45.6)	30 (30.9)	122 (51.7)	0.001 ^a^
Too expensive	14 (4.2)	5 (5.2)	9 (3.8)	0.600 ^a^

Abbreviations: SABA, Short Acting Beta Agonist; * No rhinitis symptoms versus rhinitis symptoms. ^a^ Participants were able to select multiple answers.

**Table 4 pharmacy-11-00115-t004:** Attitudes and perceptions about asthma (n = 333).

Agreement to Attitude Items, n (%)	Overall (n = 333)	No Rhinitis Symptoms (n = 97)	Rhinitis Symptoms (n = 236)	*p* Value *
I have no time to think about my health as other things are more important	107 (32.1)	19 (19.6)	88 (37.3)	0.002 ^a^
I see myself as healthy and fit	236 (70.9)	69 (71.1)	167 (70.8)	1.000 ^a^
If someone asked me, I would say I had a stressful life	141 (42.3)	39 (40.2)	102 (43.2)	0.628 ^a^
I worry about what my asthma will be like in 10 years	152 (45.6)	38 (39.2)	114 (48.3)	0.147 ^a^
My symptoms are not serious	182 (54.7)	64 (66.0)	118 (50.0)	0.008 ^a^
I find my inhaler difficult to use	31 (9.3)	6 (6.2)	25 (10.6)	0.299 ^a^
I feel embarrassed carrying my asthma inhaler around with me	61 (18.3)	17 (17.5)	44 (18.6)	0.877 ^a^
I feel embarrassed using my asthma inhaler in front of others	93 (27.9)	23 (23.7)	70 (29.7)	0.286 ^a^
My asthma stops me living life to the full	134 (40.2)	37 (38.1)	97 (41.1)	0.712 ^a^
My asthma affects my sex life	78 (23.4)	21 (21.6)	57 (24.2)	0.671 ^a^

* No rhinitis symptoms vs. rhinitis symptoms; ^a^ Participants were able to select multiple answers.

## Data Availability

Data are available in a public, open access repository. Data are available upon reasonable request.

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
