# Peer review of "Uncovering the Burden of Rhinitis in Patients Purchasing Nonprescription Short-Acting β-Agonist (SABA) in the Community"

_pharmacy, 2023, doi:10.3390/pharmacy11040115_

Round 1

Reviewer 1 Report

This is interesting study of patients purchasing SABA in a country where SABA is sold over the counter and without prescription. This study highlights the role of the pharmacy personnel in advising asthma and rhinitis patient to appropriate medication use meaning both controller medication for asthma and topical glucocorticoids for rhinitis symptoms instead of using only SABA.

MAJOR

I didn’t find comments on the Ethical Board process on this study. Did this study have an agreement from Ethical Board? If not what is the explanation?

Was the questionnaire validated or was it based on some other previous questionnaires or was it made by the Authors?

Those with rhinitis symptoms had gastroeosophageal reflux symptoms twice as often than those who did not report rhinitis symptoms. Obesity did not differ between the two groups. What could be the explanation for the association between rhinitis symptoms and GERD? Please, discuss.

There are two interesting subgroups in this study. The first one is those asthma patients that have only SABA as their asthma medication regardless of their prescriptions. The second one is those asthma patients having moderate to severe rhinitis symptoms. None of these groups are considered in the analyses. Especially the first one (SABA as only asthma medication) is important from the public health perspective. I would very much recommend further analysis on this group (if these were men or women, old or young, had hay fever or not, had had asthma exacerbation or oral glucocorticoids in the past 12 months etc).

Reviewer 2 Report

The manuscript by Alamyar and co-workers addresses the prevalence of self-reported rhinitis symptoms in patients with asthma purchasing SABA OCT in community pharmacies in New South Wales (Australia) and differences in medication-related behavioral characteristics. A cross-sectional study design was applied. Main data source consisted of structured, self-administrated questionnaires. Although the methodology has its limitations, the results are interesting for the readership of the journal as they underline the large group of asthma patients in daily pharmacy practice with suboptimal asthma control (with and without rhinitis symptoms) that would require additional support.

Main comments:

1.     Primary outcome definition: although the question to assess self-reported rhinitis symptoms was derived from previously used definitions, it is unclear whether it has been previously validated. It does not include a timeframe and only a quarter of the patients reported taking rhinitis medication. Is there more information available to strengthen the outcome definition: e.g. whether the rhinitis was doctor-diagnosed and whether rhinitis medication was prescribed? Were the FeNO levels higher in the group with self-reported rhinitis? The appendix does show asthma questionnaires, but questions on rhinitis are not included, therefore it is unclear which information on rhinitis was collected.

2.     SABA OTC: please add a paragraph in the introduction explaining the context of SABA OTC in Australia, since this is not common practice in all countries.

3.     Has it been documented whether the SABA OTC concerns refills / first prescriptions? Did results differ between those 2 groups?

4.     Previous work shows that older asthma patients are more often adherent to maintenance treatment then younger patients. Was there an age-effect regarding the medication-related behaviors?

5.     Study population: What was the response rate of the study? Were there differences in the characteristics of responders and non-responders? Could selection bias have influenced the results?  

6.     Data analysis: non-parametric tests were not performed: did all variables follow a normal distribution?

7.     Data analysis: the analyses mainly address whether there are differences in the characteristics of patients with and without rhinitis symptoms; in addition it would be interesting to assess whether rhinitis is an independent predictor of poor asthma outcomes / SABA overuse in this population when correcting for other risk factors, for example by performing logistic regression.  

8.     Title: please include SABA use (/OTC) in the title, to indicate that the study has been performed in a specific subgroup of asthma patients.    

Minor comments:

-       Table 1: please separate current and past smokers.

-       Conclusions: what kind of interventions would be of interest here? 
